# Anaerobic photoinduced Cu(0/I)-mediated Glaser coupling in a radical pathway

Siqi Zhang[1,2] & Liang Zhao [1] ✉

The reaction mechanism of the historic copper-catalyzed Glaser coupling has been debated to be based on redox cycles of Cu ions in specific oxidation states or on a radical mechanism based on Cu(0)/Cu(I). Here, the authors demonstrate two coexisting Glaser coupling pathways which can be differentiated by anaerobic/irradiation or aerobic reaction conditions. Without $O_2$, copper(I) acetylides undergo a photo-excited pathway to generate highly reactive alkynyl radicals, which combine together to form a homo-coupling product or individually react with diverse X-H (X = C, N, O, S and P) substrates via hydrogen atom transfer. With $O_2$, copper(I) acetylides are oxidized to become a Cu-acetylide/Cu-O merged Cu(I/II) intermediate for further oxidative coupling. This work not only complements the radical mechanism for Glaser coupling, but also provides a mild way to access highly energetic alkynyl radicals for efficient organic transformations.

Glaser coupling reaction of terminal alkynes is a classical textbook reaction with over 150-year history[1]. It provides an indispensable and easily accessible synthetic approach to efficiently construct 1,3-diynes as an important building block in polymers[2–4], macrocycles[5–7] and rigid supramolecular architectures[8–10]. Starting from the initial catalyst version in 1869 by using CuCl and $NH_4OH$ under $O_2$ atmosphere[11], it has evolved from the combination of Cu(II)-acetate and pyridine to the most popular CuCl-TMEDA(*N,N,N′,N′*-tetramethylethylenediamine) adduct established by Hay and co-workers[12]. Continuous mechanistic studies have revealed that Cu(I) acetylides as a key intermediate probably experience oxidation by $O_2$ to generate a dimeric Cu(II) acetylide species[13–15] towards the final oxidative coupling step (Fig. 1a). Nevertheless, the composition and structure of catalytically active Cu(I)-acetylide intermediates remains elusive because synthetic Cu(I) acetylides are easily aggregated into insoluble polymers with low kinetic activity in chemical reactions. Lately, a Cu(I)-Cu(II) synergistic model was proposed as an alternative intermediate based on X-ray absorption spectroscopy investigations[16]. In addition, Cu(II)-alkynyl species[17–20] are also considered as key intermediates in both Glaser coupling and other copper-catalyzed organic transformations[21–23], which readily disproportionate by other Cu(II) complexes to generate high valence Cu(III) species for the completion of the oxidative

coupling as a result of the Cu(I)-Cu(II)-Cu(III) redox cycle (Fig. 1b). On the other hand, a radical-involved mechanism in terms of the oxidation of acetylide by Cu(II) has been proposed at the very early stage of Glaser coupling in 1936[24,25]. However, it was subsequently repudiated in view of harsh conditions for the generation of alkynyl radicals (C(sp)-H bond dissociation energy ≈ 130 kcal mol⁻¹) and their short life time[26–29]. To date, the detailed mechanism of the Glaser coupling is still a largely unsolved problem, primarily owing to the complexity of variable copper oxidation states and a diverse range of possible copper-oxygen intermediates.

In this work, we demonstrate two different Glaser coupling pathways under anaerobic and aerobic conditions by means of a macrocycle-encircled octanuclear Cu(I)-acetylide cluster $[(PhC≡C)_4Cu^I_8(MeCN)_4]$@**Py[8]** (**1**) (**Py[8]** = azacalix[8]pyridine). In the absence of $O_2$ and under the irradiation of visible light, **1** undergoes a homo-coupling pathway as similar as the common Cu(I)/$O_2$-based Glaser reaction. The radical trapping and EPR monitoring experiments validate the in situ generation of highly reactive alkynyl radicals upon photoexcitation (Fig. 1c). Moreover, the energetic Ph-C ≡ C˙ exhibits high hydrogen atom transfer (HAT) reactivity towards diverse X-H (X = C, N, O, S, and P) substrates. In the presence of $O_2$, **1** could be oxidized to form a Cu-acetylide/Cu-O merged Cu(I/II) cluster $[(PhC≡CCu^{II}Cu^I)-(μ_2-O)-Cu^{II}]$@**Py[8]** (**2**) as the first

[1]Key Laboratory of Bioorganic Phosphorus Chemistry & Chemical Biology (Ministry of Education), Department of Chemistry, Tsinghua University, Beijing 100084, China. [2]Jiangsu Key Laboratory of New Drug Research and Clinical Pharmacy, Xuzhou Medical University, Xuzhou, Jiangsu 221004, China. ✉e-mail: zhaolchem@mail.tsinghua.edu.cn

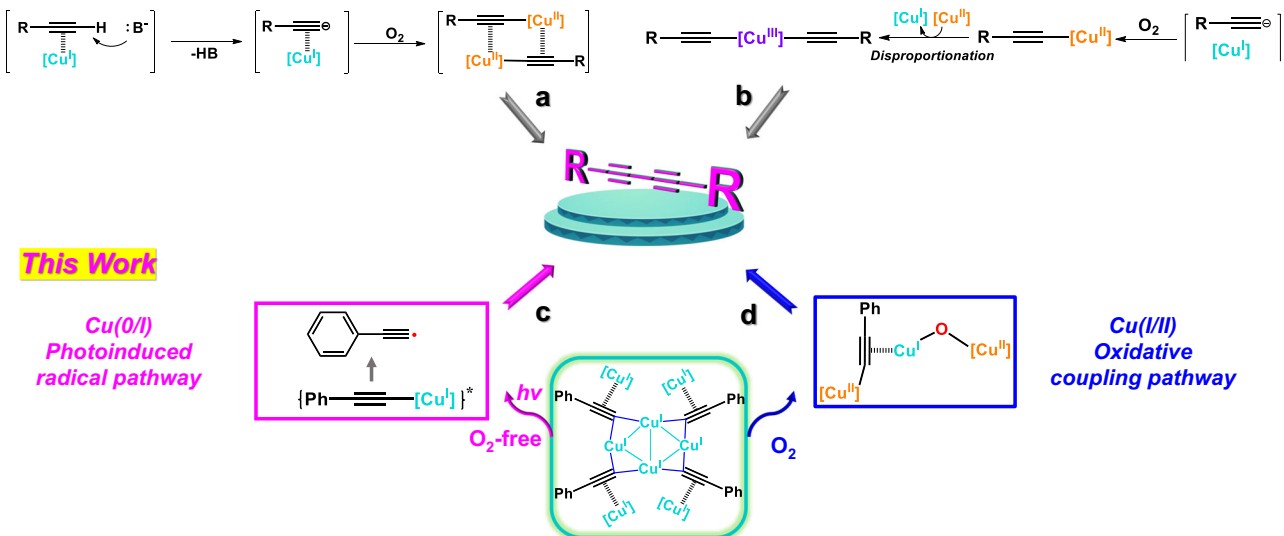

**Fig. 1 | Distinct proposed mechanisms for the Glaser coupling. a** Dimeric Cu(II) acetylides as key intermediates. **b** Disproportionation of Cu(II)-alkynyl species. **c** Anaerobic photoinduced coupling mediated by Cu(0/I) in a radical pathway.

**d** Aerobic oxidative coupling mediated by a merged Cu(II) acetylide and Cu-O species.

isolated mixed-valence bicopper acetylide intermediate. The release of the merged Cu(I/II) cluster core out from the macrocycle leads to the homo-coupling product as well (Fig. 1d). Thus, the anaerobic/photo-excitation and aerobic conditions distinguish two feasible coupling pathways of the historic Glaser reaction. The unprecedented radical mechanism via a $O_2$-free photo-induced Cu(0/I)-mediated process not only provides a mild way to access highly energetic alkynyl radicals for efficient organic transformations, but also exemplifies a new mechanistic perspective for copper-catalyzed alkynylation and photo-induced transformations[29,30].

## Results

### Macrocycle-encircled Cu(I)-acetylide cluster 1

Cu(I) acetylides are supposed as key intermediates in many $Cu/O_2$ catalyzed coupling reactions. However, their structures vary with the substituted groups on terminal alkynes from polymeric, oligomeric to discrete clusters[31,32]. Therein, the polymeric (e.g., $[(PhC≡CCu^I)_\infty]$ as a chain polymer) or oligomeric (e.g., the interlocking ring-like $[(^tBuC≡CCu^I)_{20}]$) copper acetylides show almost no kinetic activity due to its extremely poor solubility. In a practical reaction system, it is expected that low nuclearity soluble Cu(I)-acetylide clusters derived from insoluble crude copper acetylides actually act as true catalytically active species, while their composition and dynamic structural variation remain elusive. We herein attempt to destruct the polymeric copper acetylides and stabilize active Cu(I)-acetylide cluster segments by coordinative and protective macrocyclic ligands. The anaerobic treatment of Ph-C ≡ C-TMS with $[Cu(CH_3CN)_4](BF_4)$ in the presence of the macrocyclic ligand azacalix[8]pyridine (**Py[8]**), which is composed of eight 2,6-pyridine rings annularly bridged by eight $NCH_3$ groups, led to a yellow crystalline compound. Its chemical formula was characterized by X-ray crystallographic analysis as $[Cu_8(\mu_3-PhC≡C)_4(CH_3CN)_4(Py[8])](BF_4)_4·3(CH_3OH)·(H_2O)$ (**1**). Therein, the cluster moiety within **Py[8]** is properly described as four identical $[PhC≡CCu^I_3]$ units fused by sharing four central Cu(I) atoms (Fig. 2a). The phenylacetylide group in each $[PhC≡CCu^I_3]$ unit adopt a mixed σ/π $\mu_3$-$\eta^1,\eta^1,\eta^2$ mode to interact with three Cu(I) atoms. This rhombus-like $Cu_4$ plane in **1** shows high geometric and dimensional similarity with the repeating unit of the previously reported polymeric $[(PhC≡CCu^I)_\infty]$ determined by X-ray powder diffraction[32,33].

Further structural characterizations by electron-spray ionization mass spectroscopy (ESI-MS), X-ray photoelectron spectroscopy (XPS), and electron paramagnetic resonance (EPR) spectra identified the composition and oxidation states of copper ions in **1**. The XPS measurement (Supplementary Fig. 1) shows only one intensive peak at 932.8 eV for Cu 2p under X-ray irradiation of 1486.6 eV. In combination with the Cu LMM Auger spectrum at 571.2 eV, the corresponding Auger parameter is deduced to be 1848.0 eV, which is close to the standard Cu(I) sample CuCl (1847.6 eV)[34]. Therefore, the copper ions in **1** are assigned as 1+ oxidation state, which is also verified by the EPR silent feature (Supplementary Fig. 2) and a high-resolution ESI-MS peak at m/z = 2021.0278 corresponding to $([Cu^I_8(PhC≡C)_4(Py[8])(BF_4)_3]^+$, Supplementary Fig. 3).

### Anaerobic photoinduced Glaser coupling of 1 in a radical pathway

Under the irradiation of visible light at 432 nm, the solid of **1** exhibited an intensive green emission at $\lambda_{em} = 530$ nm (Supplementary Fig. 4). However, the photo-excitation of **1** in solution resulted in quick quenched luminescence even under $N_2$ atmosphere (Fig. 2b), which was accompanied with the solution color change from yellow to dark orange. TEM monitoring revealed an irradiation-caused stepwise aggregation from nanoclusters (d ≈ 5 nm) to large nanoparticles (d ≈ 150 nm, Fig. 2c and Supplementary Fig. 5). This process was also validated by a series of mass spectra peaks with escalating molecular weights (Supplementary Fig. 6). Furthermore, GC-MS monitoring on the photo-irradiated solution identified the formation of the homo-coupling PhC≡C-C≡CPh as a major organic product (Supplementary Fig. 7). In order to make clear the possible redox cycle in the photo-irradiated transformation, we conducted XPS measurement on the isolated product, which showed no characteristic shake-up peak of Cu(II) (Supplementary Fig. 8). Further peak fitting illustrated two intensive peaks at 932.3 and 932.8 eV for Cu 2p in a ratio of 2:1 under X-ray irradiation of 1486.6 eV. In combination with the Cu LMM peaks at 570.6 and 568.1 eV respectively, the corresponding Auger parameters were deduced as 1848.3 and 1851.3 eV, which are in good consistence with the standard 1+ sample CuCl (1848.0 eV) and metallic Cu(0) (1851.3 eV). In addition, the EPR silence result also confirmed no Cu(II) formed during this photo-induced coupling reaction (Supplementary Fig. 9).

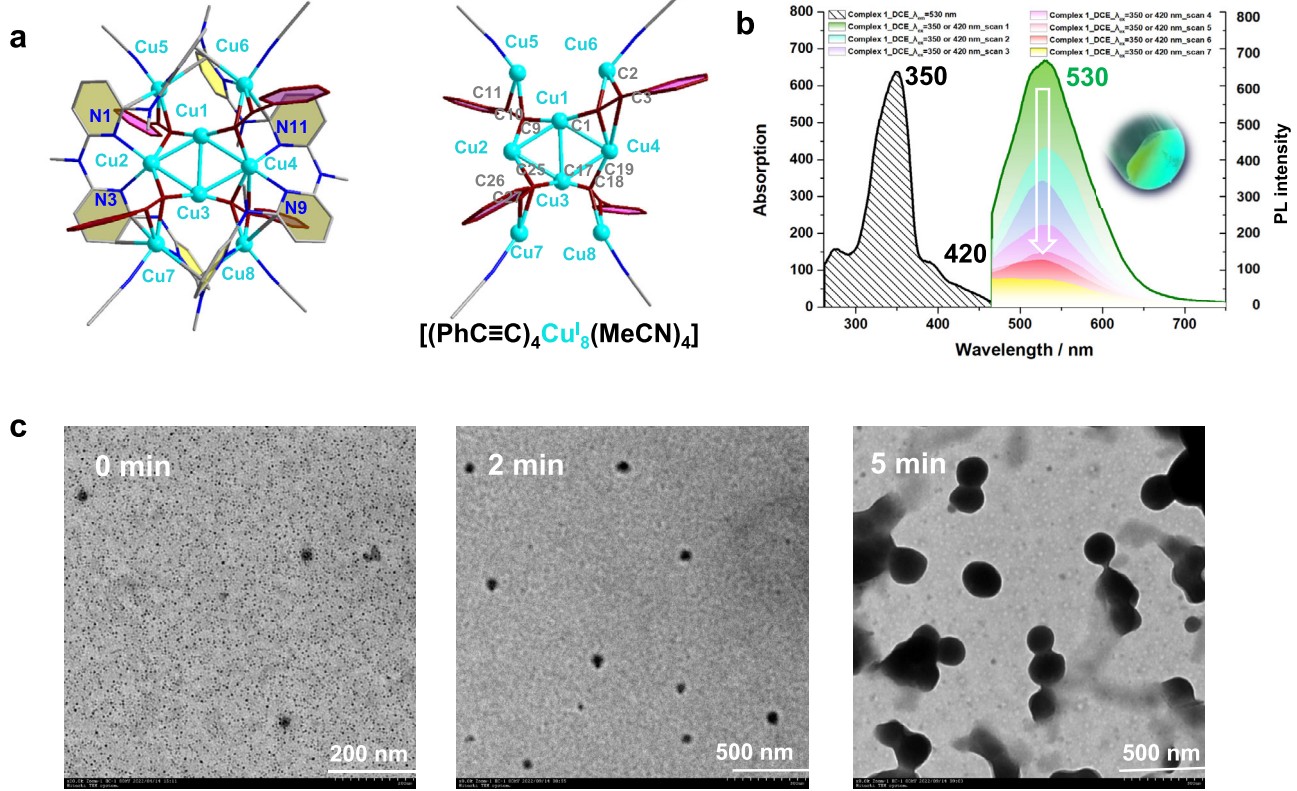

**Fig. 2 | Structural characterization of 1. a** Crystal structure and the cluster core of **1** with partial atom labeling (50% thermal ellipsoid probability level). Hydrogen atoms, $BF_4^-$ anions and solvent molecules are omitted for clarity. **b** Absorption and emission spectra of **1** in dichloroethane (DCE) with different scan times. **c** TEM images of **1** under the irradiation of $\lambda_{ex} = 420$ nm at different intervals.

We next tried to gain detailed mechanistic insights into this Cu(0/I)-mediated Glaser type coupling under the anaerobic/photo-induced condition. Firstly, a cross reaction of **1** and its F-substituted analogue **1-F** ([(4-F-$C_6H_4$-C≡C)$_4$Cu$^I_8$(MeCN)$_4$]@**Py[8]**) under irradiation was applied to make sure that the coupling reaction is intra- or inter-molecular. The identification of both homo- and hetero-coupling products substantiates its inter-molecular pathway (Supplementary Fig. 10). In combination with the *vide supra* Cu(0)/Cu(I) redox cycle and a considerable yield of 25% for the hetero-coupling product 4-F-$C_6H_4$-C≡C-C≡C-Ph, we postulate a possible radical-based reaction pathway that includes the homolysis split of RC≡C-Cu(I) to generate RC≡C· and the further combination of two alkynyl radicals outside the **Py[8]** macrocycle to form either homo- or hetero-coupling products (Fig. 3a). This radical-based Glaser type coupling pathway has been rarely paid attention due to the difficult generation of alkynyl radicals (C(sp)-H bond strength ≈ 130 kcal mol⁻¹), which generally requires a very high reaction temperature.

The generation of highly energetic phenylethynyl radicals in the photo-induced Glaser type coupling of **1** was confirmed by radical trapping experiments. The employment of TEMPO[28] or 1,1-diphenylethylene[35,36] yielded the radical trapping adducts **3** and **4** as evidenced by HR-MS (Fig. 3b and Supplementary Fig. 12). A similar radical trapping experiment of **1-F** also resulted in two corresponding radical trapping adducts **3-F** and **4-F** (Supplementary Fig. 13). Moreover, EPR spin-trapping experiments with DMPO (5,5-dimethyl-1-pyrroline N-oxide) offer a more direct approach to identify the phenylethynyl radical[28]. The irradiation on the mixed sample [**1** + DMPO] showed intensive EPR signals that can be assigned as a carbon species-based radical **5** ($A_{Hβ} = 22$ G; $A_N = 15$ G) based on simulation (Fig. 3c). In the control experiments of [**1** + DMPO] in the dark

(Fig. 3c) or just the DMPO under irradiation, no EPR signals were detected (Supplementary Fig. 14).

The energetic alkynyl radicals in situ generated by the photo-induced transformation of **1** showed high reactivity with diverse X-H substrates including alkynes, amines, alcohols, thiols, and phosphines. As a result of hydrogen atom transfer (HAT) between the active Ph-C≡C· and the X-H substrates, a series of desired radicals (MeO·, HOO·, $C_6H_{11}$S·, $H_2$N·, Ph(N·)H and $Ph_2$P·) were identified by EPR in the irradiated [**1**+substrates] samples with DMPO as the capture agent (Fig. 4). Taking [**1** + NH$_3$•H$_2$O] as an example, the simulation for the resulting EPR signal reveals a typical amino radical with $A_{Hβ} = 15.85$ G, $A_N = 19.03$ G and $A_{Nβ} = 1.70$ G (theoretical values: $A_{Hβ} = 15.85$ G, $A_N = 19.03$ G and $A_{Nβ} = 1.71$ G, Supplementary Fig. 15), which is generally formed by exciting the ammonia gas at a high temperature. Furthermore, we also applied diyne substrates to verify the feasibility of radical transfer. As evidenced by HR-MS, a diversity of oligomer products via multi-step coupling were identified (Fig. 4 and Supplementary Fig. 16). The homo-coupling product of the diyne substrate itself provides further evidence for the photo-induced radical transfer mechanism.

Furthermore, we found that the O$_2$-free photo-induced Cu(0/I)-mediated Glaser coupling pathway is also applicable to the polymeric [(PhC≡CCu)$_∞$] (**6**). Despite of the poor solubility of **6**, the corresponding radical trapping adduct **5** was also generated either in [**6** + DMPO] (Fig. 3c) or [**6** + DMPO] with **Py[8]** or the commonly used auxiliary ligands such as TMEDA and 1,10-phenanthroline (Supplementary Fig. 17). Moreover, the radical transfer of the newly generated Ph-C≡C· towards X-H (X = N, O, S, and P) substrates via HAT also worked very well (Supplementary Fig. 18). These results collectively substantiate that this novel alkynyl radical pathway under anaerobic and photo-induced condition is universal in the common Glaser coupling reaction.

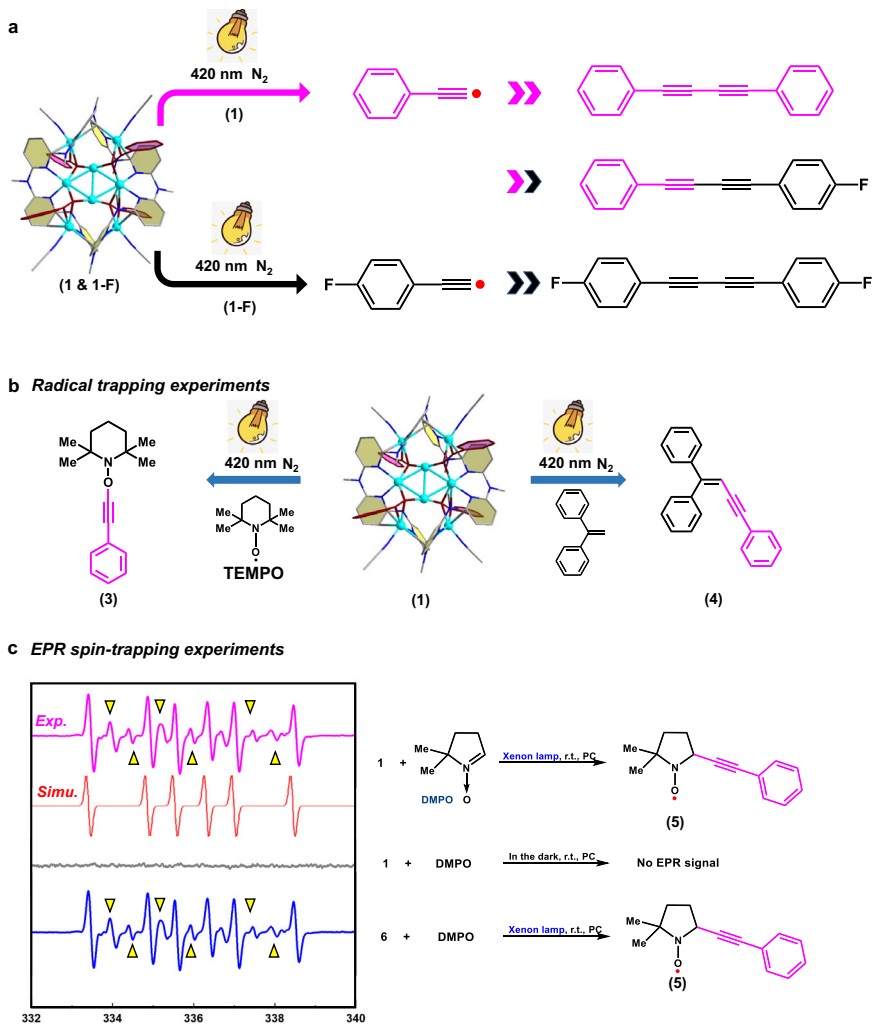

**Fig. 3 | Photo-induced radical mechanism and radical transfer from Ph-C≡C· in 1.**
**a** Cross experiment of **1** and **1-F** in propylene carbonate (PC) under the irradiation at 420 nm in N₂ atmosphere for 30 min. **b** Radical trapping experiments for **1** in PC under irradiation with different radical capture agents (left: TEMPO; right: 1,1-diphenylethylene) for 10 min. **c** Experimental and simulated EPR spectra of **1** and **6** with DMPO under in situ irradiation with Xenon lamp (filter >360 nm) in N₂ atmosphere. EPR signals with triangle are assigned as the oxidized product of DMPO. Simulations are listed in Supplementary Fig. 11.

**Aerobic Glaser coupling mediated by a merged Cu(I/II) species 2**

Upon exposure to O₂, **1** was oxidized to yield a merged cluster [Cu₃(μ₂-PhC≡C)(μ₂-O)(CH₃CN)(**Py[8]**)](BF₄)₂·(CH₂Cl₂) (**2**) (Fig. 5a). It is notable that the binuclear [PhC≡CCu₂] unit in **2** has essential difference with the previously reported Cu(I) acetylide structures[31–33]. The σ-bonded copper atom Cu3 (Cu-C: 1.908(8) Å) exhibits a typical square planar coordination geometry of Cu(II). Detailed EPR and XPS measurements confirm the mixed Cu^I/Cu^II oxidation states of copper ions in **2**. The EPR signature for **2** shows a characteristic *g* value for the square planar coordinated Cu^II center (g∥ (g_z = 2.185) > g⊥ (g_{x(y)} = 2.082, g_{y(x)} = 2.000), Supplementary Fig. 19). The shake-up peaks in XPS also support the existence of Cu^II ions in **2**[37] (Supplementary Fig. 20). In combination with the XPS peak fitting result and the appearance of two Cu^II/I reduction peaks in a cyclic voltammetry (CV) experiment (Supplementary Fig. 21), the ratio of Cu^I and Cu^II was assigned as 1:2. In addition, the crystallographic analysis and ESI-MS characterization (Supplementary Fig. 22) confirm the composition of **2** as a μ₂-oxo bridged species [(PhC≡CCu^IICu^I)-O-Cu^II]. This unique structure includes the characteristics of many previously proposed intermediates for the Glaser coupling, such as the Cu(I)-Cu(II) synergism[16], the Cu(II) alkynyl[17–20], and the merged Cu-acetylide/Cu-O intermediate[38]. Natural bond orbital (NBO) analysis reveals that the

existence of such unique Cu(II) alkynyl moiety may arise from the balanced charge distribution on both Cu(II) (0.800 for Cu1; 0.779 for Cu3) and Cu(I) (0.728 for Cu2) (Fig. 5b).

The release of the merged [(PhC≡CCu^IICu^I)-(μ₂-O)-Cu^II] core from **Py[8]** by heating or adding coordinative solvents (e.g., CH₃CN) led to the homo-coupling diyne product as well (Supplementary Fig. 23). Meanwhile, along with the addition of CH₃CN, the EPR signature for Cu(II) in **2** disappeared immediately (Fig. 5c), which should be attributed to the Cu(II)-mediated reductive elimination of phenylacetylides in the merged [(PhC≡CCu^IICu^I)-(μ₂-O)-Cu^II] core. In contrast, no coupling reactions took place if irradiating **2** by visible light (Supplementary Fig. 24). Clearly, two different pathways including the photo-induced Cu(0/I)-mediated radical coupling of **1** and the synergistic Cu(I/II)-mediated oxidative coupling of **2** co-exist in the macrocycle-based copper acetylide system, and they are differentiated by anaerobic/photo-excitation or aerobic conditions.

## Discussion

Cu/O₂-catalyzed C-C and C-heteroatom coupling reactions have been a continuance attention hotspot ever[39]. To date, a number of important copper-oxygen or organo-copper complexes with high absorption of visible light have been identified as active intermediates[39,40], which

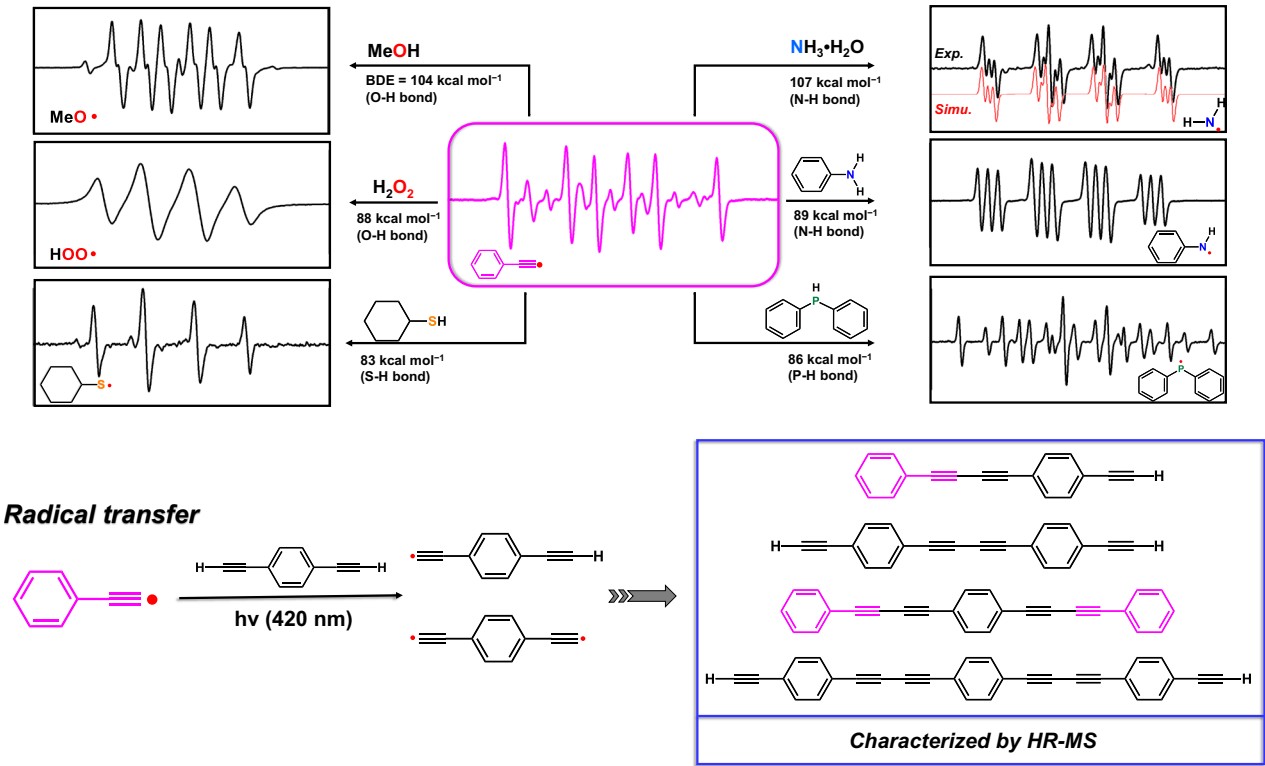

**Fig. 4 | Radical transfer between Ph-C ≡ C˙ and various X-H substrates.** Experimental and simulated EPR spectra of **1** plus X-H substrates with DMPO in PC under in situ irradiation (Xenon lamp, filter >360 nm) in N₂ atmosphere. BDE is the abbreviation of bond dissociation energy. EPR simulations are listed in Supplementary Fig. 15. HR-MS spectra for the oligomerized products in PC under the irradiation at 420 nm in N₂ atmosphere for 30 min are shown in Supplementary Fig. 16.

exhibits multifarious redox reactivity by a one-electron (e.g., $Cu^I/Cu^{II}$) or two-electron mode (e.g., $Cu^I/Cu^{III}$) due to rich oxidation states of copper ions. However, the effect of visible light irradiation has not been fully taken into account. This work demonstrates two coexisting Glaser acetylenic coupling pathways under anaerobic/irradiation or aerobic reaction conditions based on the macrocycle-encircled Cu(I)-acetylide cluster **1**. In the absence of $O_2$, the visible light irradiation on **1** efficiently generates highly energetic alkynyl radicals, which promote the Glaser acetylenic coupling and hold a high capacity of reacting with diverse X-H (X = C, N, O, S, and P) substrates through HAT process. At this point, we reveal the indispensable effect of visible light in $Cu/O_2$-catalyzed transformations, and provide an easily accessible strategy for accessing the highly energetic alkynyl radicals. In the other aspect, the Cu(I)-acetylide cluster **1** could be oxidized by $O_2$ to form the Cu-acetylide/Cu-O merged Cu(I/II) cluster **2** in the same Glaser coupling reaction system. The release of the merged [(PhC≡$CCu^{II}Cu^I$)-(μ₂-O)-$Cu^{II}$] core leads to the homo-coupling of two phenylacetylides in a common oxidative coupling way. As a consequence, we expand the library of active copper-oxygen species in $Cu/O_2$-catalyzed alkynylation transformations from Cu(I)- or Cu(II)-acetylide through the Cu(I)-Cu(II) synergistic motif to this mixed valence copper(I/II)-oxo-acetylide merged model[21–23].

In conclusion, we have revealed two distinct coupling pathways, involving an anaerobic photo-induced radical coupling and an aerobic oxidative coupling, in the same Glaser coupling system. The highly energetic alkynyl radicals show remarkable HAT reactivity with diverse X-H substrates (X = C, N, O, S, and P), potentiating as a considerable radical source in multifarious radical transfer processes. In addition, the isolated Cu-acetylide/Cu-O merged Cu(I/II) cluster [(PhC≡$CCu^{II}Cu^I$)-(μ₂-O)-$Cu^{II}$]@**Py[8]** represents the first μ₂-η¹,η² bimetallic acetylide intermediate for the Glaser coupling. Unveiling two coexisting reaction pathways in the same reaction system may provide a new mechanistic perspective on many Cu-catalyzed organometallic transformations under anaerobic or aerobic conditions. The easy access of highly energetic alkynyl radicals by visible light irradiation may offer a facile way to initiate alkynyl radical-based organic transformations and polymerizations towards functional materials.

## Methods
### General Information

All commercially available chemicals were used without further purification. The solvents used in this study were processed by standard procedures. Mass spectra were obtained using a Thermo Scientific Exactive Orbitrap instrument. UV-vis measurements were performed using Agilent Cary 7000 UV-Vis-NIR spectrophotometer. EPR experiments were carried out using JEOL JES-FA200 ESR Spectrometer and CIQTEK EPR100. Transmission electron microscopy (TEM) measurements were performed on Hitachi H-7650 microscope. Luminescent spectra were recorded on an Edinburgh spectrofluorimeter (FLS980). Luminescent decay experiments were measured on an Edinburgh FLS980 spectrometer using time-correlated single photon counting (TCSPC). The details of X-ray crystallographic measurements are summarized in X-ray Crystallographic Analysis. Octamethylazacalix[8] pyridine (**Py[8]**) macrocycle was synthesized according to the reported synthetic protocol[41] by Pd-catalyed [3 + 5] fragment coupling reactions between a terminal dibrominated linear pentamer and a terminal diaminated linear trimer.

### Synthesis of complex 1 ([Cu₈(μ₃-PhC ≡ C)₄(CH₃CN)₄(Py[8])] (BF₄)₄·3(CH₃OH)·(H₂O))

Under nitrogen protection, **Py[8]** (8.5 mg, 0.01 mmol) and [Cu(CH₃CN)₄](BF₄) (31.4 mg, 0.10 mmol) were dissolved in a mixed solvent of anhydrous methanol and dichloromethane (1.5 mL, v/v = 1/1) in a 10 mL Schlenk tube. Then Ph-C ≡ C-TMS (10 μL) was injected and

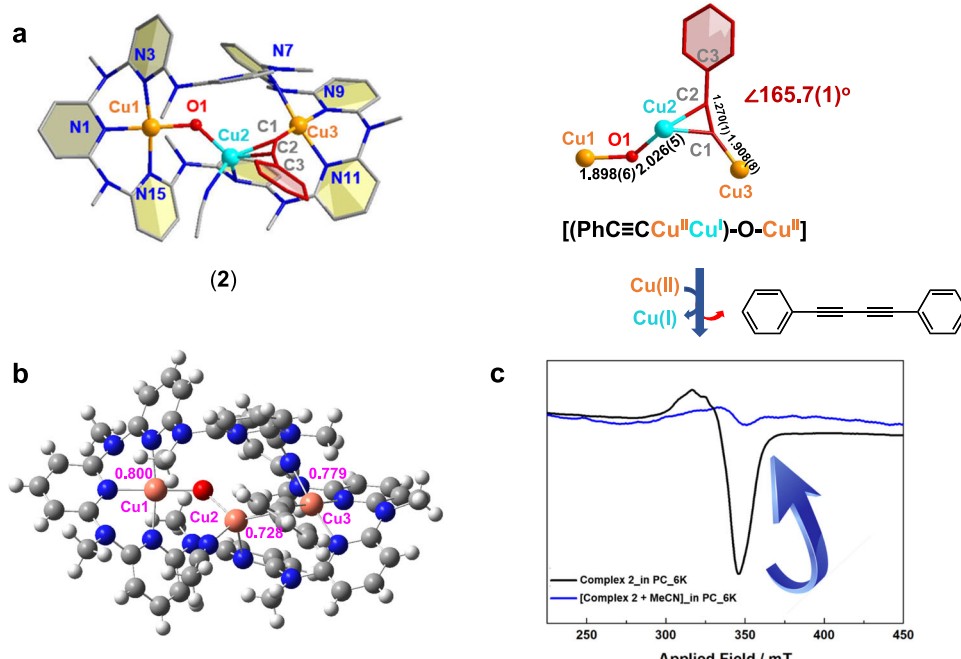

**Fig. 5 | Structural characterization of 2. a** Crystal structure and the cluster core of **2** with partial atom labeling (50% thermal ellipsoid probability level). Hydrogen atoms, $BF_4^-$ anions, and solvent molecules are omitted for clarity. **b** NBO analysis of **2. c** EPR monitoring on the occurrence of Glaser coupling of **2** before (black) and after (blue) the addition of coordinative solvent MeCN (6 K, 9.04 GHz).

the mixture was stirred for 3 h until the solution became turbid. After filtration, light yellow crystals of **1** were obtained in 41% yield (8.2 mg) by one-day evaporation. High-resolution ESI-MS: m/z = 2021.0271 for $[Cu^I_8(PhC\equiv C)_4(Py[8])(BF_4)_3]^+$.

### Synthesis of complex 1-F ($[Cu_8(4\text{-}F\text{-}C_6H_4\text{-}C\equiv C)_4(CH_3CN)_4(Py[8])](BF_4)_4$)

The synthetic procedure for **1-F** was similar to that of **1** but using 4-F-$C_6H_4$-C ≡ C-TMS in place of Ph-C ≡ C-TMS. Light yellow crystals of **1-F** were obtained in 45% yield (9.5 mg) by one-day evaporation. However, the crystal was too sensitive and unstable to be structurally characterized by XRD. High-resolution ESI-MS: m/z = 2092.5035 for $[Cu^I_8(4\text{-}F\text{-}C_6H_4\text{-}C\equiv C)_4(Py[8])(BF_4)_3]^+$.

### Synthesis of complex 2 ($[Cu_3(\mu_2\text{-}PhC\equiv C)(\mu_2\text{-}O)(CH_3CN)(Py[8])](BF_4)_2\cdot(CH_2Cl_2)$)

Complex **2** was isolated from the same reaction system of **1** by exposure to air and extending the evaporation time. Brown crystals of **2** were obtained in 20% yield (2.7 mg) by three-day evaporation. High-resolution ESI-MS: m/z = 1331.2590, 622.1277, and 385.7505, corresponding to $[Cu^ICu^{II}_2(PhC\equiv C)(O)(Py[8])(BF_4)_2(H)]^+$, $[Cu^ICu^{II}_2(PhC\equiv C)(O)(Py[8])(BF_4)(H)]^{2+}$ and $[Cu^ICu^{II}_2(PhC\equiv C)(O)(Py[8])(H)]^{3+}$, respectively.

### Computational details

Theoretical calculations of **2** were performed using the Gaussian 09 program[42]. Initial structures for natural bond orbital (NBO) analysis of **2** for calculations were built up on the basis of single-crystal structure. Becke three parameter hybrid functional accompanied by Lee–Yang–Parr correlation functional (B3LYP)[43,44] was employed in DFT calculation without any symmetry constraints on molecular structures. Dunning correlation-consistent basis set ccpVTZ-pp (a triple-ζ basis set)[45,46] was applied for copper atoms and 6-311 G** basis set[47] for other atoms in NBO calculation. The Hay and Wadt effective core potentials with a double-ζ basis set (LanL2DZ)[48–51] were applied for copper atoms and 6-31 G basis set[47] for other atoms in optimization for molecular orbitals. The root is set as 1 in all of the DFT calculations.

## Data availability

All data generated in this study are provided in the Supplementary Information/Source data file. Full characterization data including high-resolution ESI-MS, EPR, XPS, UV-vis spectra, CV and experimental details are listed in the supplementary information. Coordinates of the optimized structures are provided as source data and can be accessed via figshare under the https://doi.org/10.6084/m9.figshare.22769495. The X-ray crystallographic coordinates for structures reported in this study have been deposited at the Cambridge Crystallographic Data Centre (CCDC), under deposition numbers 2252404 (**1**) and 2252405 (**2**). These data can be obtained free of charge from The Cambridge Crystallographic Data Centre via www.ccdc.cam.ac.uk/data_request/cif. All other data are available from the corresponding author upon request. Source data are provided with this paper.

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

## Acknowledgements

Financial supports by NSFC (22025105 and 21821001) and Shuimu Tsinghua Scholar Program are gratefully acknowledged.

## Author contributions

L.Z. and S.Z. conceived and supervised the project. The synthetic experiments and structural characterizations were carried out by S.Z. All authors discussed the results and commented on the manuscript.

## Competing interests

The authors declare no competing interests.
