## [Peer Review File · Nature Communications]

Anaerobic photoinduced Cu(0/I)-mediated Glaser coupling in a radical pathwayReviewers' Comments:

Reviewer #1:

Remarks to the Author:

In this manuscript, Dr. Zhao and co-workers have reported that the Glaser coupling reaction mechanism involves coexisting pathways differentiated by aerobic or anaerobic/irradiation conditions, with the former leading to oxidative coupling and the latter generating highly reactive alkynyl radicals for efficient organic transformation.

The practical implications of this paper are significant as it provides a new mechanistic perspective for copper-catalyzed alkynylation and photoinduced transformations. This work also complements the radical mechanism for Glaser coupling, which can be useful in the development of new catalysts for organic synthesis. The easy access of highly energetic alkynyl radicals by visible light irradiation may offer a facile way to initiate alkynyl radical-based organic transformations and polymerizations.

The paper uses various methods, such as radical trapping experiments, HR-MS spectra, EPR spectroscopy, cyclic voltammetry, and X-ray crystallography to investigate the reaction mechanism of the copper-catalyzed Glaser coupling under two different conditions. The authors also used density functional theory (DFT) calculations to support their proposed mechanism.

I recommend that this work is suitable for publication in Nature Communication Journal after including these minor revisions.

1. For easy-to-understand and reproducibility reasons, the authors should add detailed information about the experiment. For example, in Fig 3 and 4, also in SI, what was the solvent and reaction time in these experiments? This information could be added to the picture or the caption. And why the high energy and short lifetime alkynyl radical could exist in this system and did not react with the solvent? In Fig 5b, what is the software and function of the NBO analysis, could it be shown in the manuscript or SI?

2. The last sentence on page 11, "Natural bond orbital (NBO) analysis reveals that the existence of such unique Cu(II) alkynyl moiety may arise from the balanced charge distribution on both Cu(II) (0.800 for Cu1; 0.779 for Cu3) and Cu(I) (0.728 for Cu2) (Fig. 5b).", it mentioned that "0.800 for Cu1; 0.779 for Cu3", however, the Fig 5b showed that "0.779 for Cu1; 0.800 for Cu3". Which one is right?

3. Please provides clear pictures in the manuscript and supporting information, for example, Fig. 2b and Fig. 5b.

Reviewer #2:

Remarks to the Author:

In this manuscript, Zhao and coauthor reported the mechanistic investigation of copper catalyzed Glaser coupling. With the success on obtaining copper-alkyne cluster complex 1 and 2, the authors were capable to explore the reaction with these complexes. EPR, XPS and TEM technologies were applied in monitoring the reactivity. Based on the results, the authors reached to the conclusions that the copper catalyzed Glaser coupling undergo to different reaction paths, depending on whether the reaction were exposed to molecular oxygen. The authors did a solid job in conducting the stoichiometric reactions with these copper clusters and were able to obtain some valid information of copper catalyzed alkyne coupling. The X-ray structures of these two copper clusters provide interesting structural information's of copper alkyne interactions. The radical quenching and photo-promoted complex "decomposition" gave some plausible explanation of what type of copper intermediates are essential for the observed alkyne coupling (formation of conjugated diyne). Overall, authors did a good job in evaluating the reactivity of these two copper clusters reaction under photo initiation. However, this reviewer felt that the reported work, although has some interesting new discovery, is not up to the bar for the expected high impact in synthesis and organometallic chemistry and could not recommend its publication for Nature Communication.

First, Glaser coupling for the synthesis of conjugated diyne is a well-studied reaction. Although there

are some debates on reaction mechanism as authors discussed in the introduction, the overall transformation is not considered as major challenge for synthesis, especially for the homo diyne coupling. Therefore, the synthetic value of this work is not high.

Second, while authors monitoring the reaction with the copper complexes is the most direct strategy in organometallic investigation, the complexity of this system makes it hard to generalize into other copper-promoted chemistry, which therefore reduced the interest of this investigation in advancing copper catalysis. Moreover, the fact that authors obtained the X-ray structures of the complexes, it suggested that these complexes are reasonably stable. The reported investigations might miss the formation of other type of high-energy intermediates and/or transition state, which contribute to the observed coupling products. After all, the reaction mechanism information more depends on kinetic study, which is very difficult for copper as multiple oxidation states and various interconvertible clusters could all attribute to the reaction path. A minor concern was that no kinetic profile (first order reaction of copper or different). As there is no reaction rate study, it is hard to draw the conclusion of the final reaction path.

Finally, one important goal for mechanistic studies is to apply the obtained information for new/important/challenging transformations. It is not clear that the formation of alkyne radical and copper- σ/π bis coordination would provide new insight information to warrant/initiate new ideas. As the homo coupling is still the targeting product, there are no obvious applications of this finding in advanced copper catalysis.

This reviewer applauds the authors for conducting a solid study. However, the questionable claim (no kinetic profile), lack of obvious applications (even assuming the proposed two reaction paths are correct) and not clear new mechanistic insight beyond current understanding, make this reviewer could not recommend the work for publication in Nature communication.

We thank the reviewers for their insightful and helpful suggestions and have made the following changes to the manuscript as suggested and requested.

Reviewer #1 (Remarks to the Author):

In this manuscript, Dr. Zhao and co-workers have reported that the Glaser coupling reaction mechanism involves coexisting pathways differentiated by aerobic or anaerobic/irradiation conditions, with the former leading to oxidative coupling and the latter generating highly reactive alkynyl radicals for efficient organic transformation. The practical implications of this paper are significant as it provides a new mechanistic perspective for copper-catalyzed alkylation and photoinduced transformations. This work also complements the radical mechanism for Glaser coupling, which can be useful in the development of new catalysts for organic synthesis. The easy access of highly energetic alkynyl radicals by visible light irradiation may offer a facile way to initiate alkynyl radical-based organic transformations and polymerizations.

The paper uses various methods, such as radical trapping experiments, HR-MS spectra, EPR spectroscopy, cyclic voltammetry, and X-ray crystallography to investigate the reaction mechanism of the copper-catalyzed Glaser coupling under two different conditions. The authors also used density functional theory (DFT) calculations to support their proposed mechanism.

I recommend that this work is suitable for publication in Nature Communication Journal after including these minor revisions.

Detailed remarks of Reviewer #1:

1. For easy-to-understand and reproducibility reasons, the authors should add detailed information about the experiment. For example, in Fig 3 and 4, also in SI, what was the solvent and reaction time in these experiments? This information could be added to the picture or the caption.

Response: We thank this Reviewer for this suggestion. Accordingly, the experimental details, including the solvent and reaction time, have been added in the figure caption of the revised manuscript and Supporting Information (SI).

2. Why the high energy and short lifetime alkynyl radical could exist in this system and did not react with the solvent?

Response: We thank this Reviewer for this point. The highly energetic alkynyl radical is generated by an *in situ* photo-induced reaction under O₂-free condition, and could only be characterized with the assistance of radical capture agents (DMPO, TMPO or 1,1-diphenylethylene) due to its short lifetime and high reactivity. The high energy of Ph-C≡C• indeed facilitates efficient reactivity with diverse X-H (X = C, N, O, S and P) substrates through hydrogen atom transfer (HAT) reactions. In the radical characterization experiments, propylene carbonate is selected as an inert solvent without active hydrogen source, making it not react with the Ph-C≡C• in the system.

3. In Fig 5b, what is the software and function of the NBO analysis, could it be shown

in the manuscript or SI?

Response: We thank this Reviewer for this comment. The software and function of the NBO analysis was shown in the manuscript on page 19. The details are listed below.

Computational details: Theoretical calculations of **2** were performed using the Gaussian 09 program. Initial structures for natural bond orbital (NBO) analysis of **2** for calculations were built up on the basis of single-crystal structure. Becke three parameter hybrid functional accompanied by Lee–Yang–Parr correlation functional (B3LYP) was employed in DFT calculation without any symmetry constraints on molecular structures. Dunning correlation-consistent basis set ccpVTZ-pp (a triple- ζ basis set) was applied for copper atoms and 6-311G** basis set for other atoms in NBO calculation. The Hay and Wadt effective core potentials with a double- ζ basis set (LanL2DZ) were applied for copper atoms and 6-31G basis set for other atoms in optimization for molecular orbitals. The root is set as 1 in all of the DFT calculations.

4. The last sentence on page 11, “Natural bond orbital (NBO) analysis reveals that the existence of such unique Cu(II) alkynyl moiety may arise from the balanced charge distribution on both Cu(II) (0.800 for Cu1; 0.779 for Cu3) and Cu(I) (0.728 for Cu2) (Fig. 5b).”, it mentioned that “0.800 for Cu1; 0.779 for Cu3”, however, the Fig 5b showed that “0.779 for Cu1; 0.800 for Cu3”. Which one is right?

Response: We thank this Reviewer for pointing out this very important issue. We have checked the NBO analysis results, and the charge distributions on Cu1 and Cu3 are assigned to be 0.800 and 0.779, respectively. The corresponding data for Fig. 5b have been corrected in the revised version.

5. Please provides clear pictures in the manuscript and supporting information, for example, Fig. 2b and Fig. 5b.

Response: We thank this Reviewer for this comment. The figures in the revised manuscript and supporting information have been updated with high resolution version.

Reviewer #2 (Remarks to the Author):

In this manuscript, Zhao and coauthor reported the mechanistic investigation of copper catalyzed Glaser coupling. With the success on obtaining copper-alkyne cluster complex **1** and **2**, the authors were capable to explore the reaction with these complexes. EPR, XPS and TEM technologies were applied in monitoring the reactivity. Based on the results, the authors reached to the conclusions that the copper catalyzed Glaser coupling undergo to different reaction paths, depending on whether the reaction were exposed to molecular oxygen. The authors did a solid job in conducting the stoichiometric reactions with these copper clusters and were able to obtain some valid information of copper catalyzed alkyne coupling. The X-ray structures of these two copper clusters provide interesting structural information's of copper alkyne interactions. The radical quenching and photo-promoted complex

“decomposition” gave some plausible explanation of what type of copper intermediates are essential for the observed alkyne coupling (formation of conjugated diyne). Overall, authors did a good job in evaluating the reactivity of these two copper clusters reaction under photo initiation. However, this reviewer felt that the reported work, although has some interesting new discovery, is not up to the bar for the expected high impact in synthesis and organometallic chemistry and could not recommend its publication for Nature Communication.

Detailed remarks of Reviewer #2:

1. First, Glaser coupling for the synthesis of conjugated diyne is a well-studied reaction. Although there are some debates on reaction mechanism as authors discussed in the introduction, the overall transformation is not considered as major challenge for synthesis, especially for the homo diyne coupling. Therefore, the synthetic value of this work is not high.

Response: We thank this Reviewer for this comment. We herein provide more details to clarify the significance of this work. We hope with the explanation and clarification, the reviewer could reevaluate this work and agree the publication of this manuscript on Nature Comm.

(1). *New mechanistic insights for Glaser coupling reaction.*

As mentioned by the Reviewer that the Glaser coupling is a very efficient way to construct diyne compounds, this historic reaction plays an important role in organic synthesis. However, its detailed mechanism is still in disputation on account of rich oxidation states of copper ions and *in situ* formed complicated Cu-acetylide/Cu-O species. Particularly, the radical-involved mechanism was proposed at the very early stage but has not been identified ever due to the high energy and short life time of alkynyl radicals. This work not only confirms the feasibility for the formation of alkynyl radicals in copper cluster species and complements the radical mechanism for the Glaser coupling, but also clarifies the function of O₂ and reveals two co-existing coupling pathways for the first time. We believe that this mechanism studies not only promote our comprehension on the Glaser coupling reaction, but also shed light on many copper-catalyzed organic transformations in terms of the single electron-mediated pathway.

(2). *Structural identification of catalytically active Cu(I)-acetylide intermediates.*

Cu(I) acetylides are commonly considered as key intermediates in many organic transformations (e.g. Sonogashira coupling, CuAAC click reactions), but the composition of catalytically active Cu(I)-acetylide intermediates remains elusive due to the poor solubility of synthetic Cu(I) acetylides. In view of the macrocyclic effect, this work successfully captures the possible Cu(I)-acetylide cluster (Cu₈(μ₃-PhC≡C)₄) by using a **Py[8]** macrocycle. The Cu₈ cluster shows high geometric and dimensional similarity with the repeating unit of the previously reported polymeric [(PhC≡CCu^I)_∞] determined by X-ray powder diffraction (Chui, S. S. Y., Ng, M. F. Y. & Che, C.-M. Structure determination of homoleptic AuI, AgI, and CuI aryl/alkylethyne)

coordination polymers by X-ray powder diffraction. *Chem. Eur. J.* **11**, 1739-1749 (2005).). In addition, we found this copper acetylide intermediates have similar radical-based coupling reaction no matter what ligands are employed.

(3). Crystal structure support for the Cu(I)-Cu(II) synergistic model.

Cu(I)-Cu(II) synergistic model for the Glaser coupling has been raised several years before based on X-ray absorption spectroscopy investigations (ref.: Bai, R. et al. Cu(II)-Cu(I) synergistic cooperation to lead the alkyne C-H activation. *J. Am. Chem. Soc.* 2014, **136**, 16760). The originality of this pathway is the formation of a mixed-valent Cu(I)-Cu(II) intermediate, which is responsible for the following dimeric and final coupling process. Even though this Cu(I)-Cu(II) intermediate was carefully characterized by X-ray absorption spectroscopy investigations, related Cu(I)-Cu(II) structures suitable for X-ray diffraction (XRD) are still a challenge. With the assistance of the Py[8] macrocycle, the mixed-valent Cu(I)-Cu(II) cluster **(2)** could be successfully isolated due to the well-known macrocyclic effect and the protective role of macrocycle. Notably, the $[\text{PhC}\equiv\text{CCu}^{\text{II}}\text{Cu}^{\text{I}}]$ moiety in **2** shows high similarity with the previously proposed Cu(I)-Cu(II) intermediate and represents the first single crystal structure evidence for the Cu(I)-Cu(II) synergistic model.

(4). The first isolated structure with most of the proposed intermediate characters.

The unique structure of **2** includes the characteristics of many previously proposed intermediates for the Glaser coupling, such as the Cu(I)-Cu(II) synergism¹, the Cu(II) alkynyl²⁻⁵ and the merged Cu-acetylide/Cu-O intermediate⁶. The release of the merged $[(\text{PhC}\equiv\text{CCu}^{\text{I}}\text{Cu}^{\text{II}})-(\mu_2\text{-O})-\text{Cu}^{\text{II}}]$ core from **Py[8]** by heating or adding coordinative solvents (e.g. CH_3CN) led to the homo-coupling diyne product, indicating the intermediate role of this merged $[(\text{PhC}\equiv\text{CCu}^{\text{I}}\text{Cu}^{\text{II}})-(\mu_2\text{-O})-\text{Cu}^{\text{II}}]$. Meanwhile, along with the addition of CH_3CN , the EPR signature for Cu(II) in **2** disappeared immediately, which should be attributed to the Cu(II)-mediated

reductive elimination of phenylacetylides in the merged $[(\text{PhC}\equiv\text{CCu}^{\text{I}}\text{Cu}^{\text{II}})-(\mu_2\text{-O})-\text{Cu}^{\text{II}}]$ core and is consistent with the previously reported Cu(II)-mediated Glaser coupling mechanisms.

Reference:

- 1) Bai, R. et al. Cu(II)-Cu(I) synergistic cooperation to lead the alkyne C-H activation. *J. Am. Chem. Soc.* **136**, 16760-16763 (2014).
- 2) Bakhoda, A. et al. Three-coordinate copper(II) alkynyl complex in C-C bond formation: the sesquicentennial of the Glaser Coupling. *J. Am. Chem. Soc.* **142**, 18483-18490 (2020).
- 3) Ziegler, M. S., Lakshmi, K. V. & Tilley, T. D. Dicopper Cu(I)Cu(I) and Cu(I)Cu(II) complexes in copper-catalyzed azide-alkyne cycloaddition. *J. Am. Chem. Soc.* **139**, 5378-5386 (2017).
- 4) Zhang, Q. et al. Radical reactivity, catalysis, and reaction mechanism of arylcopper(II) compounds: the missing link in organocopper chemistry. *J. Am. Chem. Soc.* **141**, 18341-18348 (2019).
- 5) Kundu, S. et al. Three-coordinate copper(II) aryls: key intermediates in C-O bond formation. *J. Am. Chem. Soc.* **139**, 9112-9115 (2017).
- 6) Zhang, S. & Zhao, L. A merged copper(I/II) cluster isolated from Glaser coupling. *Nat. Commun.* **10**, 4848 (2019).

(5). New access for high-active alkynyl radicals in an efficient and mild condition.

Alkynyl radicals with high energy (C(sp)-H bond dissociation energy $\approx 130 \text{ kcal mol}^{-1}$) have attracted much recent attention due to their high efficiency in hydrogen atom transfer (HAT) with diverse substrates and the corresponding wide applications in radical transfer processes (ref: Xie, J. et al. A highly efficient gold-catalyzed photoredox $\alpha\text{-C}(\text{sp}^3)\text{-H}$ alkylation of tertiary aliphatic amines with sunlight. *Angew. Chem. Int. Ed.* 2015, **54**, 6046). However, the highly energetic alkynyl radicals could only be generated in very harsh conditions due to their short life time¹⁻³. This work provides a mild way to access highly energetic alkynyl radicals under visible light irradiation at room temperature, which shows highly efficient HAT reactivity with many X-H (X = C, N, O, S and P) substrates and could be potentially applied in diverse organic transformations.

2. Second, while authors monitoring the reaction with the copper complexes is the most direct strategy in organometallic investigation, the complexity of this system makes it hard to generalized into other copper promoted chemistry, which therefore

reduced the interest of this investigation in advancing copper catalysis.

Response: We thank this Reviewer for this point. Cu-catalyzed reactions (e.g. Glaser coupling, Sonogashira reaction, Huisgen cycloaddition, and Cadiot-Chodkiewicz coupling, etc) are a significant research field in organometallic transformations. However, their mechanistic research remains a challenge due to rich oxidation states of copper ions and complicated Cu-acetylide/Cu-O species. With the assistance of Py[8] macrocycle, we not only isolate the Cu(I)-acetylide active species, but also demonstrate the alkynyl radical using radical trapping experiments based on EPR and HR-MS. In view of this work and previously reported photo-induced copper catalyzed transformations, we believe that this radical-involved mechanism shows a good universality in other copper-catalyzed catalysis (Creutz, S. E., Lotito, K. J., Fu, G. C. & Peters, J. C. Photoinduced Ullmann C-N coupling: demonstrating the viability of a radical pathway. *Science*, 2012, **338**, 647; Zhang, L. et al. Photoinduced inverse Sonogashira coupling reaction. *Chem. Sci.* 2022, **13**, 7475). In particular, we found that the O₂-free photo-induced Cu(0/I)-mediated Glaser coupling pathway is also applicable to the polymeric [(PhC≡CCu)_∞] in the presence of the commonly used auxiliary ligands such as TMEDA and 1,10-phenanthroline. These results collectively substantiate that this novel alkynyl radical pathway under anaerobic and photo-induced condition is universal in the common Glaser coupling reaction.

3. Moreover, the fact that authors obtained the X-ray structures of the complexes, it suggested that these complexes are reasonable stable. The reported investigations might miss the formation of other type of high-energy intermediates and or transition state, which contribute to the observed coupling products.

Response: We thank this Reviewer for this comment. Due to the macrocyclic effect and protection role of Py[8] macrocycle, the copper clusters [(PhC≡C)₄Cu^I₈(MeCN)₄]@Py[8] (**1**) and [(PhC≡CCu^ICu^{II})-(μ₂-O)-Cu^{II}]@Py[8] (**2**) are stable in solid state and can be successfully isolated for XRD characterization. Nevertheless, when the central copper clusters are released out from Py[8], they are very active species to promote the coupling process. For example, the release of the merged [(PhC≡CCu^ICu^{II})-(μ₂-O)-Cu^{II}] core from Py[8] in complex **2** by heating or adding coordinative solvents (e.g. CH₃CN) promptly led to the homo-coupling diyne product and the disappeared EPR signature for Cu(II), which is attributed to the Cu(II)-mediated reductive elimination of phenylacetylides in the merged [(PhC≡CCu^ICu^{II})-(μ₂-O)-Cu^{II}] core. Therefore, the macrocycle Py[8] in these two systems mainly function as an reservoir, which could stabilize the high-energy copper clusters in solid for XRD characterization and reinstate their reactivity upon Py[8] release.

4. After all, the reaction mechanism information in more depends on kinetic study, which is very difficult for copper as multiple oxidation states and various interconvertible clusters could all attribute to the reaction path. A minor concern was that no kinetic profile (first order reaction of copper or different). As there in no reaction rate study, it is hard to draw the conclusion of the final reaction path.

Response: We thank this Reviewer for this point. Due to the high reactivity and short lifetime of the active species ($\text{Ph-C}\equiv\text{C}^\bullet$ and the merged Cu(I/II) species), the mechanistic studies in this work are indeed very complicated and no unique and characteristic peaks in UV-vis absorption or fluorescence spectra are suitable for monitoring. Thus, it is difficult to carry out the corresponding kinetic study.

However, other detailed characterizations were systematically conducted to identify the possible reactive species in the two reported reaction pathways in this work. For complex **1**, the Cu(I/0) reaction cycle and the formation of Cu(0/I) nanoclusters was confirmed by time-dependent EPR, XPS and further TEM. Besides, the radical trapping experiments based on EPR and HR-MS evidenced the formation of $\text{Ph-C}\equiv\text{C}^\bullet$, and demonstrated the photo-induced radical-involved mechanism in a Cu(0/I) cycle. For complex **2**, detailed research for this kind of merged Cu(I/II) intermediate was reported in our previous work (*Nat. Commun.* **10**, 4848 (2019)). The uniqueness of **2** arises from its inclusion of almost all the characteristics of previously proposed intermediates for the Glaser coupling, such as the merged Cu-acetylide/Cu-O intermediate, the Cu(II) alkynyl and the Cu(I)-Cu(II) synergism. Moreover, the release of the merged $[(\text{PhC}\equiv\text{CCu}^{\text{I}}\text{Cu}^{\text{II}})-(\mu_2\text{-O})-\text{Cu}^{\text{II}}]$ core from Py[8] by heating or adding coordinative solvents (e.g. CH_3CN) led to the homo-coupling diyne product as well. Meanwhile, along with the addition of CH_3CN , the EPR signature for Cu(II) in **2** disappeared immediately, which should be attributed to the Cu(II)-mediated reductive elimination of phenylacetylides in the merged $[(\text{PhC}\equiv\text{CCu}^{\text{I}}\text{Cu}^{\text{II}})-(\mu_2\text{-O})-\text{Cu}^{\text{II}}]$ core. Therefore, even though the kinetic study is difficult to carry out, other detailed characterizations could clearly demonstrate the photo-induced Cu(0/I)-mediated radical coupling of **1** and the synergistic Cu(I/II)-mediated oxidative coupling of **2**, which are differentiated by anaerobic/photo-excitation or aerobic conditions.

5. Finally, one important goal for mechanistic studies is to apply the obtained information for new/important/challenging transformations. It is not clear that the formation of alkyne radical and copper-sigma/pi bis coordination would provide new insight information to warrant/initiate new ideas. As the homo coupling is still the targeting product, there is no obvious applications of this finding in advance copper catalysis.

Response: We thank this Reviewer for this comment. The highly energetic alkynyl radicals ($\text{C}(\text{sp})\text{-H}$ bond dissociation energy $\approx 130 \text{ kcal mol}^{-1}$) endow them with high efficiency and wide applications in organic synthesis, such as HAT towards diverse X-H (X = C, N, O, S and P) substrates and radical transfer reactions. This work provides an efficient and mild method to achieve the highly energetic alkynyl radicals under visible light irradiation at room temperature, which could facilitate the development of new synthetic methods.

Besides, in view of the biased difficulty in the generation of diverse alkynyl radicals, cross coupling products could be achieved based on rational design of copper acetylide cluster moieties. A preliminary attempt has revealed that $R\text{-C}_6\text{H}_4\text{-C}\equiv\text{C}^\bullet$ with different electronegative substitutions show different cross coupling reactivity. As

shown in the figure below, the yields of coupling products increase along with the electronegativity variation of substitutions ($\text{F-C}_6\text{H}_4\text{-C}\equiv\text{C-C}\equiv\text{C-C}_6\text{H}_4\text{-F} > \text{F-C}_6\text{H}_4\text{-C}\equiv\text{C-C}\equiv\text{C-Ph} > \text{Ph-C}\equiv\text{C-C}\equiv\text{C-Ph} > \text{OMe-C}_6\text{H}_4\text{-C}\equiv\text{C-C}\equiv\text{C-Ph} > \text{OMe-C}_6\text{H}_4\text{-C}\equiv\text{C-C}\equiv\text{C-C}_6\text{H}_4\text{-OMe}$). It may be ascribed to the electrophilic nature of different acetylenic radicals, which make them more reactive with electronegative substitutions. This result provide a potential method to adjust the yield of homo- or hetero-coupling products. More synthetic applications are still in progress. We believe the mechanistic studies lay a solid foundation to carry out the following synthetic applications, and should be published separately.

Reviewers' Comments:

Reviewer #1:

Remarks to the Author:

Suitable for publication.

Reviewer #2:

Remarks to the Author:

In this revision, the authors worked very hard to address my previous concerns. These efforts include: 1) revision on the significance of mechanistic insights; 2) structure identification for plausible catalytically active intermediates; 3) clarification of the critical role of Py[8] for the successful isolation of potentially highly reactive intermediate; 4) listed some potential application of this radical-driven transformations. Although, this reviewer still has some reservations on the potential usefulness of this chemistry, given the good efforts and scholarly presented results, this reviewer would be ok to recommend this work for publication in Nat. Commun.

We thank the reviewers for their insightful and helpful comments and have made the following changes to address the remaining concerns of reviewers and editorial requests.

Reviewer #1 (Remarks to the Author):

Suitable for publication.

Response: We thank this Reviewer for this helpful comment. The final manuscript that suitable for publication has been uploaded accordingly.

Reviewer #2 (Remarks to the Author):

In this revision, the authors worked very hard to address my previous concerns. These efforts include: 1) revision on the significance of mechanistic insights; 2) structure identification for plausible catalytically active intermediates; 3) clarification of the critical role of Py[8] for the successful isolation of potentially highly reactive intermediate; 4) listed some potential application of this radical-driven transformations. Although, this reviewer still has some reservations on the potential usefulness of this chemistry, given the good efforts and scholarly presented results, this reviewer would be ok to recommend this work for publication in Nat. Commun.

Response: We thank this Reviewer for this insightful comment. Further study on the potential applications of this chemistry is still ongoing.